# Association Between Serum 25(OH)-Vitamin D and Heart Involvement in a Single-Centre Cohort of Children with Acute Rheumatic Fever During the Years 2004–2024

**DOI:** 10.3390/biomedicines13102502

**Published:** 2025-10-14

**Authors:** Donato Rigante, Gabriella De Rosa, Angelica Bibiana Delogu, Giulia Pignataro, Claudia Di Pangrazio, Marcello Candelli

**Affiliations:** 1Department of Life Sciences and Public Health, Fondazione Policlinico Universitario A. Gemelli IRCCS, 00168 Rome, Italy; gabriella.derosa@unicatt.it (G.D.R.); angelicabibiana.delogu@unicatt.it (A.B.D.);; 2Università Cattolica Sacro Cuore, 00168 Rome, Italy; 3Department of Emergency Anesthesiological and Reanimation Sciences, Fondazione Policlinico Universitario A. Gemelli IRCCS, 00168 Rome, Italymarcello.candelli@policlinicogemelli.it (M.C.)

**Keywords:** acute rheumatic fever, rheumatic heart disease, valvular heart disease, carditis, echocardiography, *Streptococcus pyogenes*, vitamin D, personalized medicine

## Abstract

**Background**: An aberrant immune response against *Streptococcus pyogenes* combined with yet-unraveled genetic inference can induce acute rheumatic fever (ARF), but factors determining the specific development of rheumatic heart disease (RHD) are obscure. **Objectives**: To retrospectively assess general and laboratory data at the onset of ARF in a single-centre cohort of children managed between 2004 and 2024, and to evaluate any potential relationship between serum vitamin D and the occurrence of RHD. **Patients and Methods**: Children with ARF diagnosed according to the revised Jones criteria, hospitalized and managed at the Department of Life Sciences and Public Health in our University, were considered; out of 90 eligible patients with post-streptococcal illness, 11 were not considered because they were diagnosed with post-streptococcal arthritis, while 1 was excluded due to incomplete inpatient data. A total final number of 78 consecutive children with ARF (39 males and 39 females) with a mean age of 10.6 ± 2.7 years was assessed via retrospective evaluation of medical records. Their demographic, clinical, and laboratory variables at disease onset, including C-reactive protein, anti-streptolysin-O titer, and 25-hydroxyvitamin D [25(OH)-vitamin D], were analyzed. **Results**: Sixty-six children (84.6% of the whole cohort) were found to have echocardiographic evidence of RHD. By dividing patients based on the presence of carditis, at the univariate analysis, we observed serum 25(OH)-vitamin D levels significantly lower in patients with cardiac involvement compared to those without (18 ± 6 *versus* 38 ± 8 ng/mL, *p* < 0.001). In addition, the proportion of patients with normal serum vitamin D levels was significantly higher among those without cardiac involvement (92%, *p* < 0.001). To account for any potential confounding factors, we performed a multivariate analysis using logistic regression, adjusted for sex and age, finding that 25(OH)-vitamin D levels lower than 30 ng/mL were the only variable associated with RHD (OR 27.752; 95% CI: 2.885–266.996). No relationship between vitamin D and the month of the year at diagnosis of ARF and RHD was found. **Conclusions**: Hypovitaminosis D was identified as a factor potentially associated with RHD occurrence in a single-centre cohort of children with ARF evaluated over two decades. This result may suggest that vitamin D deficiency contributes to the occurrence of carditis in ARF.

## 1. Introduction

Acute rheumatic fever (ARF) occurs as a short-term nonsuppurative sequela of pharyngitis caused by *Streptococcus pyogenes* [1]. Among Streptococci, which are classified according to the carbohydrate composition of cell wall antigens—such as polysaccharides, teichoic acids—as well as sequence variations in the *emm* gene encoding the M protein, only certain strains of group A beta-hemolytic *Streptococcus* have been linked to ARF [2]. The immune-mediated mechanisms underlying this important disease have not yet been fully elucidated, and only a small number of specific *emm*/M protein types can be considered pathogenic, i.e., potentially characterized by the possibility of inducing ARF, with differences between high- and low-income countries [3]. The most significant ARF-related complication is rheumatic heart disease (RHD), which may cause substantial effects on the individual’s health, often resulting in chronic illness and even premature death [4]. ARF and RHD cause a large worldwide burden of morbidity, but factors contributing to the specific involvement of the heart in ARF remain unclear. Indeed, RHD is still a major cause of cardiac failure in different countries due to poor rates of diagnosis and the lack of proper ARF treatment or prophylaxis [3]. Despite the widespread application of the Jones criteria, carditis may result in either underdiagnosis or sometimes overdiagnosis. However, the only strategy available for effective ARF control is secondary prophylaxis with long-acting penicillin G-benzathine administered every three weeks [4].

A definite reason for heart valve injuries occurring in ARF attacks is not understood, but 25-hydroxyvitamin D [25(OH)-vitamin D] deficiency has been associated with various cardiovascular risk factors and has been linked to a higher mortality following renin–angiotensin–aldosterone system activation, abnormal nitric oxide regulation, oxidative stress, and altered multiple inflammatory pathways [5]. Several observational studies have found low vitamin D levels in patients with cardiovascular diseases compared to healthy individuals [6].

The general aim of this retrospective study was to assess the clinical and laboratory characteristics of a single-centre cohort of patients with ARF hospitalized in our department over a 20-year period and to assess whether serum levels of vitamin D measured near the diagnosis of ARF were associated with the occurrence of RHD.

## 2. Patients and Methods

Out of 90 eligible pediatric patients in this study, all with a post-streptococcal illness, 11 were not considered because they were diagnosed with post-streptococcal arthritis, and 1 was excluded due to incomplete inpatient data. Patients considered in our evaluation were 78 and were all Caucasian, having an average age at ARF diagnosis of 10.6 ± 2.7 years (range 2.4–15 years); all of them were living in the central or southern Italy (at a latitude varying from 41°13′ N to 37°56′ N, with ultraviolet index of 1–3 in wintertime) and were managed at the Department of Life Sciences and Public Health of our University in Rome. All patients with ARF are regularly hospitalized in our context, and no case of suspected ARF is managed on an outpatient basis. At the time of the first clinical assessment, a questionnaire focusing on the eventual exposure to risk factors during the four weeks prior to illness, along with a general interview of the patients’ parents, was administered.

Diagnosis of ARF was established according to the modified Jones criteria, designed to identify the initial attack of ARF through a defined combination of major and minor manifestations with evidence of a previous group A *Streptococcus pyogenes* infection. Indeed, a throat swab for group A *Streptococcus pyogenes* was obtained from all patients in the cohort. The same criteria, supporting the use of Doppler echocardiography (Philips Medical, Andover, MA, USA; equipped with multifrequency S12-8, S8-3, and X5-1 transducers depending on the patient size) for the diagnosis of carditis as a major manifestation of ARF, were used to evaluate mitral and aortic valve involvement (regurgitation in at least two views, jet length ≥ 2 cm in at least one view, peak velocity > 3 m/s, confirmed via Doppler flow velocity evaluation) even in the absence of classic auscultatory findings [7].

Exclusion criteria for participation to this study were comorbidities with immunodeficiency or other immune-mediated disorders, such as systemic juvenile idiopathic arthritis and blood diseases. Demographic data, month of the year at diagnosis, clinical signs, and laboratory variables at disease onset, including C-reactive protein (CRP) and anti-streptolysin O (ASO) titer, were retrieved from medical charts. Vitamin D assessment was part of the routine workup for all hospitalized patients in our department; its serum levels, measured using automated chemiluminescence immunoassay technology (Atellica Siemens Healthineers^TM^, Swedish Hospital, Chicago, IL, USA) within the first three days of hospitalization, were retrieved from medical charts; only five patients in the cohort underwent a second blood sampling within the first two weeks post-hospitalization. All patients diagnosed with ARF underwent Doppler echocardiography to assess any cardiac involvement, and echocardiography examinations were repeated as case by case required; these patients received regular long-term follow-up (the minimal follow-up duration for a minority of cases was 5 years, while others were assessed yearly with a complete cardiologic evaluation).

## 3. Ethics Approval

The Local Ethics Committee authorized different study protocols related to nutritional issues (including vitamin D) in patients with complex diseases, as well as rare conditions and rheumatologic disorders (approval code: 2105; approval date: 5 February 2019). The whole study was performed in line with the principles of the 1964 Declaration of Helsinki and its later amendments. All patients’ caregivers were informed about the aims of this retrospective evaluation, and all of them signed a written consent for both unrestricted access to patients’ medical charts and for the evaluation of children’s anonymized data.

## 4. Statistical Analysis

Data were presented as mean (m) ± standard deviation (SD) for normally distributed continuous variables, as median (M) and interquartile range (IQR) for not-normally distributed continuous variables, and as count (percentages) for categorical variables. The Student’s *t*-test and the Mann–Whitney U test were used to compare continuous variables (for normally and non-normally distributed variables, respectively), while the chi-square test or Fisher’s exact test (depending on group size) was applied for categorical data. Additionally, we performed a multinomial logistic regression, adjusting for sex and age and including all factors that showed a *p*-value less than 0.2 in the univariate analysis. Furthermore, we constructed a receiver operating characteristic (ROC) curve with 500 bootstrap repetitions to evaluate the diagnostic accuracy of serum vitamin D levels in predicting cardiac involvement in patients with ARF. Finally, we used Youden’s test to identify the optimal cut-off value of vitamin D for distinguishing patients with and without cardiac involvement, and we also calculated the sensitivity, specificity, and positive and negative predictive values for that threshold.

## 5. Results

Among the enrolled patients, 50% were males and 50% females. Fifty patients (64%) had a positive *Streptococcus pyogenes* pharyngeal swab, while the mean ASO titer was 1433 ± 1171 IU/mL in the totality of patients (it is considered pathologically increased if >200 IU/mL). Thirty-nine patients of the cohort (50%) had arthritis in at least one joint, and the median number of involved joints was 0.5, with an interquartile range (IQR) of 0–2. Seventy percent of patients (55) reported arthralgia, and twenty-three percent (18) exhibited neurological symptoms (chorea) of varying severity. Cardiac involvement was observed in 66 patients (84.6% of the cohort). Among these, 63 patients (81%) displayed mitral regurgitation (mild in 41, moderate in 19, and severe in 3), while 45 (58%) showed aortic valve regurgitation (mild in 35, moderate in 7, and severe in 3). The severity of mitral and aortic regurgitation was assessed via echocardiography with specific metrics related to jet length ≥ 2 cm and peak velocity > 3 m/s. Of 66 patients with carditis, 24 (36.4%) had a single valve involvement, while 42 (63.6%) had both mitral and aortic valve involvement. Four patients (5%) presented signs of myocarditis, and four (5%) also had pericarditis. The most common electrocardiographic abnormality was an increased PR interval duration (first-degree atrioventricular block), which was found in 12 patients (15% of the cohort). Additionally, 15 patients (19%) showed left ventricular dilation on echocardiography, and 2 had severe heart disease requiring specific heart surgical intervention. As conceivable, a significantly higher proportion of patients with cardiac involvement required corticosteroid treatment compared to those without carditis (59% *versus* 25%, *p* = 0.05). The distribution of clinical manifestations of ARF in patients considered by this study is shown in Table 1.

By dividing patients based on the presence (or absence) of carditis, univariate analysis revealed no significant differences in age, sex, and other ARF signs (fever, arthritis, arthralgia, chorea, and/or skin lesions). Serum vitamin D levels were significantly lower in patients with ARF and cardiac involvement compared to those without (18 ± 6 *versus* 38 ± 8 ng/mL, *p* < 0.001). In addition, the proportion of patients with normal serum vitamin D levels was significantly higher in those without cardiac involvement (92%) compared to those with carditis (3%, *p* < 0.001). No differences were found between the two groups regarding the period of diagnosis. Notably, 33 patients (42% of the cohort) had their vitamin D level measured during spring or summer season (from May to October), while the remaining were tested during the wintertime. Specifically, 29 patients with carditis (44% of the cohort) were diagnosed during winter, compared to 4 patients without cardiac involvement (33.3% of the cohort, *p* = 0.54). To account for any potential confounding factors, we performed a multivariate analysis using logistic regression, adjusting for sex, age, and period of diagnosis (spring/summer or fall/winter), and including all the variables that showed a *p*-value ≤ 0.2 in the univariate analysis. The results of the logistic regression are shown in Table 2, revealing a wide confidence interval, which may be related to our model instability, likely explained by the relatively limited sample size of patients.

Serum levels of 25(OH)-vitamin D less than 30 ng/mL, assessed near the time of ARF diagnosis, were strongly associated with the presence of cardiac involvement. Concurrently, we also used vitamin D levels and the presence of cardiac involvement to generate a ROC curve with 5000 bootstrap replications to evaluate diagnostic accuracy and obtain the 95% confidence intervals (CIs). The accuracy evaluated as the area under the ROC curve was 0.965 (95% CI: 0.894–1.000), with a standard error of 0.028 and *p* < 0.001 (see Figure 1).

Therefore, we applied the Youden index, and the optimal vitamin D cut-off value for diagnosing cardiac involvement in ARF was found to be 32.5 ng/mL; this cut-off showed a sensitivity of 98.5 (95% CI: 91.84–99.96), a specificity of 91.7 (95% CI: 61.52–99.79), a positive predictive value of 98.5 (95% CI: 90.86–99.76), and a negative predictive value of 91.7 (95% CI: 60.98–98.73).

We divided patients with RHD into two groups to assess whether vitamin D levels were correlated with the severity of cardiac involvement according to echocardiography findings. The first group included patients with mild valvular involvement, while the second included those having moderate-to-severe involvement of heart valves. We chose to combine patients with moderate and moderate/severe manifestations due to the limited number of patients with severe involvement. Patients were classified into the moderate/severe group if echocardiographic findings showed at least one cardiac valve with moderate or severe regurgitation or if they had heart failure with reduced ejection fraction. All patients who did not meet these criteria were assigned to the mild cardiac involvement group. As a result, the presence of arthritis and vitamin D levels between 20 and 30 ng/dL were associated with mild RHD, whereas vitamin D levels below 20 ng/mL were linked to moderate and severe forms of RHD. Male sex and lower mean serum levels of vitamin D demonstrated a trend (not statistically significant) towards an association with moderate-to-severe carditis (see Table 3). We also performed multiple logistic regression analysis to correct these results for age, sex, and season in which ARF was diagnosed. This analysis confirmed that the presence of arthritis was associated with milder forms of RHD; conversely, vitamin D levels lower than 20 ng/mL were significantly associated with moderate and severe forms of carditis (Table 4).

## 6. Discussion

The present study reveals that vitamin D levels are statistically and independently associated with the development of cardiac involvement in patients with ARF, and that a vitamin D cut-off of 32.5 ng/mL exhibits considerable diagnostic accuracy to suggest the occurrence of RHD, with high sensitivity and specificity. Interestingly, very low levels of vitamin D (˂20 ng/mL) were associated with severe RHD.

The occurrence of RHD is common in countries with socioeconomic disadvantage, household crowding, or poor access to primary health care services as a consequence of undertreated pharyngeal infections caused by group A *Streptococcus pyogenes* [8,9]. Global estimates suggest that millions of individuals may develop RHD, with a potential risk of death above 1.4 million per year [10]. The mitral valve is most widely involved in RHD, but aortic valve involvement occurs in ¼ of cases as well [11]. A broader knowledge about risk factors and pathophysiological mechanisms underlying RHD should help optimize prophylaxis strategies for both ARF and RHD.

Indeed, improvements in living conditions and the employment of antimicrobial drugs are thought to have virtually eliminated ARF in most high-income countries [12,13]. Several aspects of nutrition may also contribute to the risk of ARF, including overall nutritional status with intake of micronutrients [14]. In particular, the serum levels of 25(OH)-vitamin D have been related to ARF in the medical literature, even though the general incidence of both streptococcal infections and ARF shows seasonal peaks during winter and spring months, when 25(OH)-vitamin D levels are expected to be lower because of the lack of sunlight exposure [15].

A host of epidemiological studies supports the association between vitamin D deficiency and the severity of different immune-mediated diseases such as ARF. For instance, poor nutritional status with hypovitaminosis D has been found in Nepalese populations, who had higher rates of RHD compared to healthy controls [16]. Yusuf et al. found significantly lower serum levels of 25(OH)-vitamin D in 55 Indian patients with RHD having mild to moderately calcified valves (median of 20.4 ng/mL) and in 55 patients with severely calcified valves (median of 11.4 ng/mL) compared to those with less severely damaged valves [17].

Vitamin D, known for its essential role in calcium and bone homeostasis, has multiple effects that go beyond the skeleton, including regulation of immunity pathways and modulation of autoimmune processes. Unfortunately, vitamin D deficiency is a global health problem, particularly in children, and several reports have revealed suboptimal serum supply of 25(OH)-vitamin D in children with IgA vasculitis, periodic fever, aphthous stomatitis, pharyngitis, cervical adenitis syndrome, and Kawasaki disease [18,19,20]. At the intersection of rheumatology and cardiovascular medicine, there is a growing awareness that recognizes how individuals with autoimmune and autoinflammatory conditions have a much higher likelihood of developing heart diseases [21]. Even children with hereditary periodic fever syndromes are prone to display heterogeneous heart complications [22,23], though the most relevant childhood condition associated with acquired cardiovascular abnormalities in children living in high-income countries is Kawasaki disease [24], requiring prompt treatment with intravenous immunoglobulin to decrease disease-associated inflammatory burden and prevent the formation of a structural damage within coronary arteries [25].

Cumulative reports show that low vitamin D status, along with genetic and environmental factors, may be involved in the pathogenesis of different immune-mediated disorders, and specifically of RHD. Onan et al. prospectively found lower levels of 25(OH)-vitamin D levels, measured with high-performance liquid chromatography, in 77% of a cohort of Turkish children with RHD compared to age-matched healthy controls having innocent murmurs over a 15-month period, with nonsignificant correlation with carditis severity [26]. We also know that the production of vascular endothelial growth factor is regulated by vitamin D, and that endothelial cell dysfunction may have a role in heart valve remodeling and in initiating the process of RHD [27,28]. Furthermore, an Arabian study identified an association between the GC2 polymorphism of the vitamin D-binding protein (VDBP) and ARF [29]. VDBP belongs to the albumin family and is essential for intracellular transport of vitamin D to various cell types, including macrophages and B lymphocytes [30]. *VDBP* gene polymorphisms have also been associated with an increased risk of mitral and aortic valve calcifications in Iranian and Mexican children with RHD [31,32].

However, our study presents several limitations. First, although patient histories were rigorously reviewed by a panel of experienced clinicians using the ARF-revised Jones criteria for diagnosis, any adjustment for potential confounding factors contributing to hypovitaminosis D, such as children’s nutritional habits, was not performed. In addition, epidemiological variations in ARF incidence were not considered, and several well-known risk factors for vitamin D deficiency in children, including limited outdoor activity (which reduces sunlight exposure) and obesity, were not accounted for. Furthermore, the retrospective design of this study, the exact timing of vitamin D assessment in relation to hospitalization, the absence of non-white children in our study, and the relatively low number of ARF cases over the past two decades (due to the single-centre nature of this study) may have influenced the final results and hindered the generalizability of our findings.

## 7. Conclusions

The development of RHD in subsets of pediatric patients with ARF is not definitively explained. This study provides a freeze image of vitamin D status at the time of ARF diagnosis, without considering specific time points. Interestingly, hypovitaminosis D was found as an independent factor associated with both RHD development and RHD severity in a single-centre cohort of children with ARF over two decades. Further work is warranted to define the role of vitamin D in ARF and RHD pathophysiology and to establish the potential existence of temporal relationships between vitamin D status and disease onset, activity, and treatment response. Such studies should involve larger cohorts of patients with a standardized assessment of potential confounding factors such as diet, sun exposure, ethnicity, and medications.

## Figures and Tables

**Figure 1 biomedicines-13-02502-f001:**
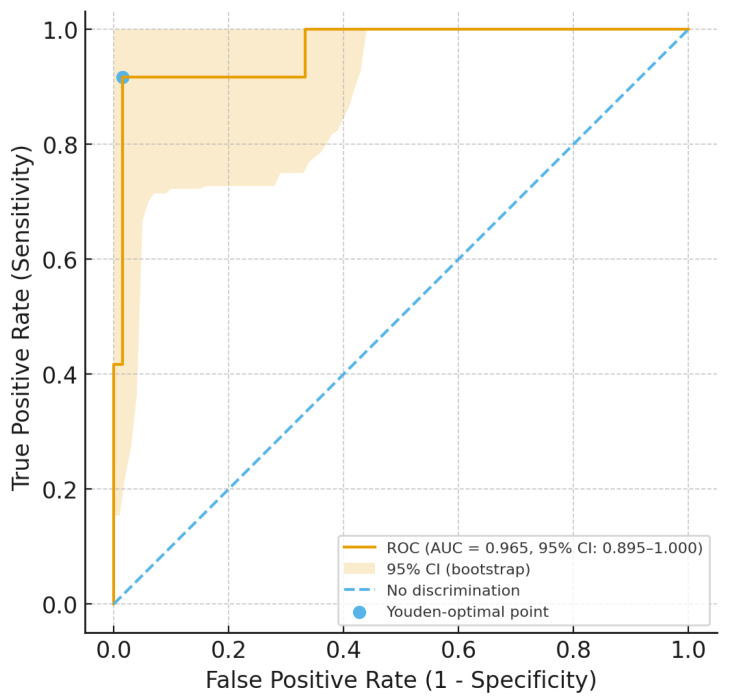
Receiver operating characteristic (ROC) curve with 5000 bootstrap replications and 95% confidence intervals (CI) for predicting cardiac involvement in patients with acute rheumatic fever based on the levels of serum vitamin D.

**Table 1 biomedicines-13-02502-t001:** Demographic, clinical, laboratory, and therapeutic data of patients diagnosed with acute rheumatic fever (ARF) in our study, stratified by the presence or absence of cardiac involvement.

	All Patients (N = 78)	ARF Patients with Cardiac Involvement (N = 66)	ARF Patients Without Cardiac Involvement (N = 12)	*p*
Age at onset (years, m ± SD)	10.6 ± 2.7	8.7 ± 2.9	8.0 ± 2.1	0.37
Males [N (%)]	39 (50)	33 (50)	6 (50)	1
Females [N (%)]	39 (50)	33 (50)	6 (50)	1
Positive *S. pyogenes* pharyngeal swab [N (%)]	50 (64)	41 (62)	9 (75)	0.52
Arthritis [N (%)]	39 (50)	32 (48)	7 (58)	0.75
Number of joints involved [M (IQR)]	0.5 (0–2)	0 (0–2)	0.5 (0–1.5)	0.52
Arthralgia [N (%)]	55 (70)	45 (68)	10 (83)	0.49
Chorea [N (%)]	18 (23.1)	13 (20)	5 (42)	0.13
Subcutaneous nodules [N (%)]	1 (1.3)	1 (1.5)	0 (0)	0.94
Erythema marginatum [N (%)]	3 (4)	3 (4.5)	0 (0)	0.91
Fever [N (%)]	43 (55)	35 (53)	8 (67)	0.53
C-reactive protein (mg/L, m ± SD)	54 ± 56	56 ± 56	43 ± 53	0.46
25(OH)-vitamin D < 20 ng/mL [N (%)]	48 (61)	47 (71)	1 (8)	0.0001
25(OH)-vitamin D between 20–30 ng/mL [N (%)]	17 (22)	17 (26)	0 (0)	0.06
25(OH)-vitamin D ˃ 30 ng/mL [N (%)]	13 (17)	2 (3)	11 (92)	0.0001
25(OH)-vitamin D (ng/mL, m ± SD)	21 ± 10	18 ± 6	38 ± 8	0.0001
Family history for ARF [N (%)]	4 (74)	4 (60)	0 (0)	0.88
Treatment with corticosteroids [N (%)]	42 (54)	39 (59)	3 (25)	0.055
Treatment with NSAIDs [N (%)]	52 (68)	47 (61)	5 (42)	0.09

ARF, acute rheumatic fever; N, number; m, mean; SD, standard deviation; M, median; IQR, interquartile range; NSAIDs, nonsteroidal anti-inflammatory drugs.

**Table 2 biomedicines-13-02502-t002:** Multivariate logistic regression analysis of factors related to cardiac involvement in patients with acute rheumatic fever (ARF) in our study after adjustment for sex, age, and season of ARF diagnosis.

	*p*	OR	95% CI
Age	0.342	0.988	0.963–1.013
Male sex	0.823	1.211	0.228–6.437
Season at ARF diagnosis (fall/winter)	0.227	0.350	0.064–1.925
Use of NSAIDs	0.680	1.420	0.268–7.514
Use of corticosteroids	0.136	3.807	0.655–22.115
Presence of chorea	0.948	1.060	0.181–6.215
25(OH)-vitamin D ˂ 30 ng/mL	0.004	27.752	2.885–266.996

OR, Odds ratio; CI, confidence interval; NSAIDs, nonsteroidal anti-inflammatory drugs.

**Table 3 biomedicines-13-02502-t003:** Demographic, clinical, laboratory, and therapeutic data of patients with rheumatic heart disease (RHD) in our study, stratified by the severity of valvular involvement.

	All Patients with RHD (N = 66)	Mild Valve Disease (N = 39)	Moderate/Severe Valve Disease (N = 27)	*p*
Age at onset (years, m ± SD)	8.7 ± 2.9	9.1 ± 2.3	8.0 ± 3.5	0.18
Males [N (%)]	33 (50)	16 (41)	17 (63)	0.08
Females [N (%)]	33 (50)	23 (59)	10 (37)	0.08
Positive *S. pyogenes* pharyngeal swab [N (%)]	41 (62)	26 (67)	15 (56)	0.44
Arthritis [N (%)]	32 (48)	24 (62)	8 (30)	0.003
Arthralgia [N (%)]	45 (68)	28 (72)	17 (63)	0.45
Chorea [N (%)]	13 (20)	8 (21)	5 (19)	0.99
Subcutaneous nodules [N (%)]	1 (1.5)	0 (0)	1 (4)	0.44
Erythema marginatum [N (%)]	3 (4.5)	1 (3)	2 (7)	0.56
Fever [N (%)]	35 (53)	20 (51)	15 (56)	0.73
C-reactive protein (mg/L, m ± SD)	56 ± 56	61 ± 62	48 ± 47	0.32
25(OH)-vitamin D < 20 ng/mL [N (%)]	47 (71)	23 (59)	24 (89)	0.008
25(OH)-vitamin D between 20–30 ng/mL [N (%)]	17 (26)	15 (38)	2 (7)	0.005
25(OH)-vitamin D ˃ 30 ng/mL [N (%)]	2 (3)	1 (3)	1 (4)	0.99
25(OH)-vitamin D (ng/mL, m ± SD)	18 ± 6	18.8 ± 6	16.0 ± 6	0.067
Treatment with corticosteroids [N (%)]	39 (59)	21 (54)	18 (67)	0.30
Treatment with NSAIDs [N (%)]	47 (61)	23 (59)	22 (81)	0.26

RHD, rheumatic heart disease; N, number; m, mean; SD, standard deviation; M, median; NSAIDs, nonsteroidal anti-inflammatory drugs.

**Table 4 biomedicines-13-02502-t004:** Multivariate logistic regression of factors related to moderate/severe rheumatic heart disease in patients with acute rheumatic fever (ARF) in our study after adjustment for sex, age, and season of ARF diagnosis.

	*p*	OR	95% CI
Male sex	0.413	0.611	0.188–1.986
Age (years)	0.322	1.009	0.992–1.026
Season at ARF diagnosis (fall/winter)	0.263	1.981	0.598–6.563
25(OH)-vitamin D ˂ 20 ng/mL	0.010	7.917	1.646–38.077
Arthritis	0.005	0.169	0.049–0.584

OR, Odds ratio; CI, confidence interval; ARF, acute rheumatic fever.

## Data Availability

No datasets were generated or analyzed during the current study.

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
