# Peer review of "Association Between Serum 25(OH)-Vitamin D and Heart Involvement in a Single-Centre Cohort of Children with Acute Rheumatic Fever During the Years 2004–2024"

_biomedicines, 2025, doi:10.3390/biomedicines13102502_

Round 1
Reviewer 1 Report
Comments and Suggestions for Authors
This retrospective study explores the association between serum 25 (OH)-vitamin D levels and the occurrence of cardiac involvement in children with acute rheumatic fever (ARF) over a 20-year period. While the manuscript addresses a clinically relevant and under-investigated topic, especially in pediatric populations, several methodological and reporting issues need to be addressed before the study can be considered for publication:
- First and foremost, the rationale for measuring 25 (OH)-vitamin D levels in these patients is not sufficiently explained. Given the retrospective nature of the study, it remains unclear why vitamin D was assessed in these children in the first place. Was vitamin D status part of a routine clinical workup for all patients, or was it measured selectively based on clinical presentation or disease severity? Without this information, the interpretation of vitamin D levels as a potential predictor of rheumatic heart disease (RHD) is susceptible to bias and confounding.
- Furthermore, the manuscript states that vitamin D levels were measured within the first two weeks after hospitalization. However, the exact timing relative to ARF diagnosis is not clearly defined. Because serum 25 (OH)-vitamin D is a dynamic biomarker influenced by acute illness, hospitalization, and seasonal variation, the lack of precise timing introduces substantial ambiguity. For the biomarker to have predictive validity regarding RHD onset or severity, its levels must be determined at or very near the time of ARF diagnosis. Any delay could result in reverse causation or dilution of the association.
- Another point that requires clarification is the laboratory methodology. While the authors mention the use of an automated chemiluminescence immunoassay system, no details are provided regarding the assay manufacturer, batch standardization, or whether all samples were analyzed using the same platform. Given the known variability among vitamin D assay techniques, particularly in retrospective multi-year studies, this information is critical for evaluating the internal validity of the results.
- The study cohort appears to be limited to hospitalized patients, although this is not explicitly stated. If only inpatients were included, this could introduce a significant selection bias. Hospitalized children are likely to represent more severe ARF cases, possibly with higher rates of cardiac involvement. Therefore, the conclusions drawn about the association between vitamin D deficiency and RHD may not be generalizable to the broader population of children with ARF, especially those managed in outpatient settings. The authors should clearly describe the inclusion criteria, particularly regarding hospitalization status, and discuss the implications of this selection bias.
- The methodology section would benefit from greater clarity in describing how ARF was diagnosed and how cardiac involvement was classified. The authors refer to the revised Jones criteria, but details on echocardiographic evaluation protocols, diagnostic thresholds for carditis, and grading of valvular involvement are lacking. Notably, the classification of RHD severity appears in the results section (rather than methods), which disrupts the logical structure of the manuscript. These definitions should be explicitly described in the methods to enhance reproducibility and transparency.
- In terms of structure and content, the manuscript would benefit from a more focused and hypothesis-driven introduction. While the authors cite literature linking vitamin D deficiency to cardiovascular and autoimmune disorders, the current introduction does not adequately build a compelling rationale for this specific study. What clinical or observational triggers prompted this investigation? What was the primary hypothesis? A clearer articulation of the knowledge gap and the intended contribution of this work would help contextualize the study's importance.
- Additionally, baseline data such as age distribution, patient numbers, and diagnostic characteristics are placed within the results section, though much of this information belongs in the methods section. Relocating these elements would improve the manuscript’s logical flow and align with conventional academic writing standards.
- The discussion appropriately recognizes several limitations, including the retrospective nature of the study, the absence of key confounding variables (e.g., sunlight exposure, dietary intake, obesity), and lack of ethnic diversity. However, the authors should more directly address the limitation introduced by the measurement timing of vitamin D and the unclear justification for its inclusion.
Author Response
Replies to REVIEWER 1
This retrospective study explores the association between serum 25 (OH)-vitamin D levels and the occurrence of cardiac involvement in children with acute rheumatic fever (ARF) over a 20-year period. While the manuscript addresses a clinically relevant and under-investigated topic, especially in pediatric populations, several methodological and reporting issues need to be addressed before the study can be considered for publication: first and foremost, the rationale for measuring 25 (OH)-vitamin D levels in these patients is not sufficiently explained. Given the retrospective nature of the study, it remains unclear why vitamin D was assessed in these children in the first place. Was vitamin D status part of a routine clinical workup for all patients, or was it measured selectively based on clinical presentation or disease severity? Without this information, the interpretation of vitamin D levels as a potential predictor of rheumatic heart disease (RHD) is susceptible to bias and confounding.
We thank the reviewer for the kind introduction given to our paper. Honestly, vitamin D determination is part of routine tests performed in children hospitalized within our department for different inflammatory conditions. This study derives from the retrospective assessment of medical charts related to children with ARF: we did not know at the start which result would have emerged from the statistical analysis. No child hospitalized had sign of rickets. No child hospitalized specifically required the investigation of vitamin D due to peculiar aspects found at the clinical assessment. At page 3 we have written that “Vitamin D assessment was part of a routine workup for all hospitalized patients in our department”.
Furthermore, the manuscript states that vitamin D levels were measured within the first two weeks after hospitalization. However, the exact timing relative to ARF diagnosis is not clearly defined. Because serum 25 (OH)-vitamin D is a dynamic biomarker influenced by acute illness, hospitalization, and seasonal variation, the lack of precise timing introduces substantial ambiguity. For the biomarker to have predictive validity regarding RHD onset or severity, its levels must be determined at or very near the time of ARF diagnosis. Any delay could result in reverse causation or dilution of the association.
Thank you for the opportunity of clarifying that vitamin D assay was mostly performed in the first two or three days of hospitalization. There were only 5 patients of the cohort in whom the test was performed at the second blood sampling, more precisely varying from day 5 to day 14 after hospital admission.
Another point that requires clarification is the laboratory methodology. While the authors mention the use of an automated chemiluminescence immunoassay system, no details are provided regarding the assay manufacturer, batch standardization, or whether all samples were analyzed using the same platform. Given the known variability among vitamin D assay techniques, particularly in retrospective multi-year studies, this information is critical for evaluating the internal validity of the results.
Thank you for this further clarification: we know that serum 25(OH)-vitamin D levels may vary according to the different methods of analysis used (chemiluminescence immunoassay, radioimmunoassay or high performance liquid chromatography). In our case the assay used was an Atellica chemistry analyzer (Siemens HealthineersTM), and we confirm that the technique was a chemiluminescence immunoassay. In the last 10 years this Atellica instrument has been and is the analyzer currently used. In previous years other analyzers (i.e. Siemens Advia Centaur) used a chemiluminescence technique as well. Unfortunately, there is no standardization of 25(OH)-vitamin D measurements among different laboratories, we are truly sorry.
The study cohort appears to be limited to hospitalized patients, although this is not explicitly stated. If only inpatients were included, this could introduce a significant selection bias. Hospitalized children are likely to represent more severe ARF cases, possibly with higher rates of cardiac involvement. Therefore, the conclusions drawn about the association between vitamin D deficiency and RHD may not be generalizable to the broader population of children with ARF, especially those managed in outpatient settings. The authors should clearly describe the inclusion criteria, particularly regarding hospitalization status, and discuss the implications of this selection bias.
In our context “all” patients with ARF are hospitalized, and no patient with a suspicion of ARF is managed on an outpatient basis. We have added in the text (at page 3) that “All patients with ARF are hospitalized in our context, and no case of suspected ARF is managed on outpatient basis”. This note should exclude any selection bias.
The methodology section would benefit from greater clarity in describing how ARF was diagnosed and how cardiac involvement was classified. The authors refer to the revised Jones criteria, but details on echocardiographic evaluation protocols, diagnostic thresholds for carditis, and grading of valvular involvement are lacking. Notably, the classification of RHD severity appears in the results section (rather than methods), which disrupts the logical structure of the manuscript. These definitions should be explicitly described in the methods to enhance reproducibility and transparency.
All patients were diagnosed with ARF according to the modified Jones’ criteria, designed to establish the initial attack of ARF through a combination of major and minor manifestations and evidence of a previous group A streptococcal infection. The reference related to such criteria is by Gewitz MH et al: Revision of the Jones Criteria for the diagnosis of acute rheumatic fever in the era of Doppler echocardiography: a scientific statement from the American Heart Association. Circulation 2015, 131, 1806-18. This important paper supports the use of Doppler echocardiography in the diagnosis of carditis as a major manifestation of ARF. We all know that echocardiography has become a cornerstone in worldwide screening programs to evaluate the prevalence of ARF and RHD. Although carditis of ARF has been considered to be a pancarditis and is clinically diagnosed, echocardiography is crucially being used to diagnose carditis and even subclinical carditis. The repetition of ultrasound heart investigations was established according to that paper (at least every 2 weeks) for all patients hospitalized with suspected ARF. The paper also highlights that it is reasonable to perform serial echocardiography/Doppler studies in any patient with diagnosis of ARF or suspected to have ARF (Class II a; Level of Evidence C). Our colleagues De Rosa G, Delogu AB and Di Pangrazio C used those criteria to confirm a pathological mitral or aortic regurgitation (pathological regurgitation in at least 2 views, jet length ≥2 cm in at least 1 view, peak velocity >3 m/s, and completion with Doppler envelope). We have added this clarification in the section of Methods, at page 3: “These same criteria, supporting the use of Doppler echocardiography in the diagnosis of carditis as a major manifestation of ARF, were used to evaluate mitral and aortic valve manifestations (regurgitation in at least 2 views, jet length ≥2 cm in at least 1 view, peak velocity >3 m/s, completion with the Doppler assessment) even in the absence of classic auscultatory findings”.
In terms of structure and content, the manuscript would benefit from a more focused and hypothesis-driven introduction. While the authors cite literature linking vitamin D deficiency to cardiovascular and autoimmune disorders, the current introduction does not adequately build a compelling rationale for this specific study. What clinical or observational triggers prompted this investigation? What was the primary hypothesis? A clearer articulation of the knowledge gap and the intended contribution of this work would help contextualize the study's importance.
Our study starts from the observation of very low vitamin D levels in children with RHD: at the start we did not have a precise goal. What we observed (after two decades) prompted us to perform this retrospective assessment. However, we have tried to better clarify the process of the whole investigation throughout the text.
Additionally, baseline data such as age distribution, patient numbers, and diagnostic characteristics are placed within the results section, though much of this information belongs in the methods section. Relocating these elements would improve the manuscript’s logical flow and align with conventional academic writing standards.
We have omitted in the section dedicated to Results the fact that patients were Caucasian and living in the central or southern Italy (at a latitude varying from 41°13’N to 37°56’N, with ultraviolet index of 1-3 in wintertime). All these features were reported in the Methods. The Results currently include the positivity rate for Streptococcus pyogenes at the pharyngeal swab, the mean anti-streptolysin O titer in patients, and the clinical manifestations of ARF which were (retrospectively) observed. The severity of mitral or aortic valvular changes was judged using echocardiography (performed by GDR, ABD, and CDP), with specific metrics including the measurement of the regurgitant volume or area and the effect of the leak on the size and function of heart chambers.
The discussion appropriately recognizes several limitations, including the retrospective nature of the study, the absence of key confounding variables (e.g., sunlight exposure, dietary intake, obesity), and lack of ethnic diversity. However, the authors should more directly address the limitation introduced by the measurement timing of vitamin D and the unclear justification for its inclusion.
The timing of vitamin D assessment has been added to the limitations of our study. As we wrote, vitamin D assessment is part of a routine workup for all hospitalized patients in our department. We have reconstructed the section dedicated to the limitations.
Reviewer 2 Report
Comments and Suggestions for Authors
Introduction
The introduction needs to be more focused. It jumps around quite a bit from varying aspects related to ARF/RHD - diagnosis, secondary prophylaxis. I think the introduction needs to introduce what ARF and RHD are. Then highlight what some of the risk factors are for developing ARF/RHD before going into the theory about how low levels of vit D might be a risk factor.
Line 40 "pharyngitides" should be pharyngitis"
Line 44 need to soften this to "has been shown to give rise to ARF" there have been studies linking other Strep strains to ARF.
The concept of "rheumatogenic" is no longer a major contributing factor to ARF.
Developing and developed countries are better to be called be low and high income.
Line 48. "most dreaded" are not quite the right words. Suggest re-wording this sentence.
RHD brings heart failure.
RHD is also a cause of death in some high-income countries.
I'm unsure what your statement about the Jones criteria means and I'm unsure how the sentence on penicillin links to the Jones criteria.
The aim needs further refining, what does "comprehensive description of the clinical and labwork characteristics" mean? As your manuscript focuses on Vit D, I suggest your aim should also.
Methods
Move the parts about who was included into the study to results. lines 71-74. Suggest using years rather than months as ages - no one can interpret how old a 15-year-old is in months.
Did the study only consider pediatric notes? Or was that only who was diagnosed?
Your methods don't actually say what was done - I.e. A retrospective evaluation of patient notes was undertaken to identify....... Included in the study were Children with ARF diagnosed according to the revised Jones criteria, all having a follow-up duration of >5 years and managed at the Department of Life Sciences and Public Health in our University, were included.
Was the questionnaire undertaken as part of the normal workup for an ARF diagnosis?
Results
Some of the parts in your results can be moved to your methods - i.e they all resided in central or Southern Italy - this must have been part of a criteria or was your study national?
Give a range for the ASO titres.
Suggest referring to the table in your results, rather than repeating what is in the table.
Tabel 3 and 4 are not needed. These results can be written in the text.
Ethics section title is needed.
Discussion
The first paragraph is not needed.
The discussion should report on your results and then link into what else is known about Vit D and cardiac involvement in patients with ARF.
Line 272 is how your results section should start. Then go into the parts you have written above.
Line 278 delete "in all truth"
Comments on the Quality of English Language
Some of the English presented is not in the right context. Suggest having an English as a first language person double check the manuscript.
Author Response
Replies to REVIEWER 2
The introduction needs to be more focused. It jumps around quite a bit from varying aspects related to ARF/RHD - diagnosis, secondary prophylaxis. I think the introduction needs to introduce what ARF and RHD are. Then highlight what some of the risk factors are for developing ARF/RHD before going into the theory about how low levels of vit D might be a risk factor.
We have revised the introduction, thank you so much for the opportunity of amending and clarifying the focus of this paper.
Line 40 "pharyngitides" should be pharyngitis"
Done.
Line 44 need to soften this to "has been shown to give rise to ARF" there have been studies linking other Strep strains to ARF.
Done, thank you for the suggestion.
The concept of "rheumatogenic" is no longer a major contributing factor to ARF.
We have omitted that adjective, thank you.
Developing and developed countries are better to be called be low and high income.
Done.
Line 48. "most dreaded" are not quite the right words. Suggest re-wording this sentence.
The sentence has been re-worded.
RHD brings heart failure.
Corrected.
RHD is also a cause of death in some high-income countries.
We have omitted the specification of countries in relationship with RHD-related deaths.
I'm unsure what your statement about the Jones criteria means and I'm unsure how the sentence on penicillin links to the Jones criteria.
You are right, there is no relationship between Jones’ criteria and long-acting penicillin prophylaxis. The sentence has been split: “Despite the widespread application of Jones' criteria, carditis may result either underdiagnosed or sometimes overdiagnosed. However, the only strategy available for effective ARF control is secondary prophylaxis with long-acting penicillin G-benzathine administered every three weeks”.
The aim needs further refining, what does "comprehensive description of the clinical and labwork characteristics" mean? As your manuscript focuses on Vit D, I suggest your aim should also.
We have clarified the general aims of our retrospective evaluation, writing that “The general aim of this retrospective study was to assess clinical and labwork characteristics of a single-centre cohort of patients with ARF hospitalized in our Department during a 20-year-period and to assess whether the serum level of vitamin D measured near to the diagnosis of ARF could be associated with occurrence of RHD”.
Methods: Move the parts about who was included into the study to results. lines 71-74. Suggest using years rather than months as ages - no one can interpret how old a 15-year-old is in months.
Changes made, thank you for the suggestion.
Did the study only consider pediatric notes? Or was that only who was diagnosed?
Our study is referred to children with ARF, as childhood is the age at which ARF is more consistently observed. We did not consider adult patients from the start.
Your methods don't actually say what was done - I.e. A retrospective evaluation of patient notes was undertaken to identify... Included in the study were Children with ARF diagnosed according to the revised Jones criteria, all having a follow-up duration of >5 years and managed at the Department of Life Sciences and Public Health in our University, were included.
The general aim of this retrospective study was to provide a description of clinical and labwork characteristics of a single-centre cohort of patients with ARF, who were hospitalized in our Department during a 20-year-period. All patients were regularly followed-up, and retrospectively analyzed to check laboratory investigations and verify if low serum vitamin D levels could be associated with the development of RHD.
Was the questionnaire undertaken as part of the normal workup for an ARF diagnosis?
Yes, it was.
Results: Some of the parts in your results can be moved to your methods - i.e they all resided in central or Southern Italy - this must have been part of a criteria or was your study national?
Done, thank you again. That’s what we observed. The fact that patients lived in central or southern Italy was not a criterion of inclusion.
Give a range for the ASO titres.
Done (the ASO titer is considered pathologically increased if >200 IU/mL).
Suggest referring to the table in your results, rather than repeating what is in the table.
We have done our best to avoid repeating concepts when tables were available.
Tabel 3 and 4 are not needed. These results can be written in the text.
We have omitted tables 3 and 4, describing the results within the text. Thank you for this suggestion.
Ethics section title is needed.
Inserted.
Discussion: The first paragraph is not needed. The discussion should report on your results and then link into what else is known about Vit D and cardiac involvement in patients with ARF.
We have reconstructed the discussion following your suggestions, thank you so much.
Line 272 is how your results section should start. Then go into the parts you have written above.
Done, thank you.
Line 278 delete "in all truth"
Done, thank you again.
Reviewer 3 Report
Comments and Suggestions for Authors
This manuscript addresses the association between serum 25(OH)-vitamin D levels and cardiac involvement in children with acute rheumatic fever (ARF). The topic is of clinical and epidemiological interest, given the ongoing burden of rheumatic heart disease (RHD) globally, especially in low- and middle-income countries. The authors present a single-center retrospective analysis spanning two decades. The findings—suggesting that hypovitaminosis D is independently associated with the development and severity of RHD—are provocative and may open avenues for preventive or adjunctive strategies.
The manuscript is clearly written and generally well-structured. However, there are several methodological limitations, interpretative overstatements, and presentation issues that need to be addressed before the work can be considered for publication. Below you will find my comments divided into major and minor.
Major
- The retrospective design and single-center nature of the study are major limitations. Although the authors acknowledge this, the conclusion occasionally overstates the causal relationship between vitamin D and RHD. The text should consistently frame the results as associations, not as evidence of causality.
- Nutritional status, obesity, outdoor activity, and socioeconomic determinants are all strong predictors of vitamin D deficiency in children. None of these variables were collected or adjusted for. Their omission raises concerns about residual confounding. This should be more thoroughly addressed in the discussion.
- The cohort includes only 78 patients over 20 years, exclusively Caucasian, from a single Italian center. This restricts external validity, particularly as ARF and RHD are more prevalent in non-Caucasian populations. The limitations in terms of generalizability should be discussed more explicitly.
- The multivariable logistic regression (Table 2) shows an extremely wide confidence interval for vitamin D (<30 ng/mL, OR ~27, CI 2.9–267). This suggests instability of the model due to the limited sample size and event distribution. The authors should discuss this limitation more carefully and avoid describing the results as “outstanding diagnostic accuracy” without nuance.
- The ROC AUC of 0.965 and a cut-off of 32.5 ng/mL seem implausibly high in a real-world clinical setting. There is a risk of overfitting. The authors should test internal validation methods (e.g., bootstrapping) or, at the very least, tone down the strength of their claims.
- All patients were said to have at least 5 years of follow-up, but some “recently diagnosed cases” were also included. This inconsistency should be clarified.
- The definition of “moderate/severe” RHD is heterogeneous (valvular regurgitation vs. heart failure). Please justify why these categories were merged and discuss how this might bias results.
- The discussion includes several citations on vitamin D and cardiovascular/autoimmune disease, but the novelty of this study relative to prior work is not sufficiently emphasized. The authors should highlight what new information is contributed compared to previous studies in Nepal, India, and the Middle East that already reported vitamin D deficiency in RHD.
Minor
- The abstract should state explicitly that this is a retrospective, single-center study, to avoid misleading readers.
- The results section of the abstract should include actual odds ratios and confidence intervals rather than only qualitative statements.
- Table 1 mixes means with medians/IQRs in a confusing way. Please use consistent reporting formats.
- Some p-values (e.g., 0.0001) should be reported as “<0.001” according to standard conventions.
- The ROC curve (Figure 1) should include the 95% CI for AUC on the figure itself.
- Several expressions are overly strong (e.g., “outstanding accuracy,” “only factor associated”). Replace with more cautious wording.
- Minor grammatical issues (e.g., “geoepidemiological variations… remains to unravel”) should be corrected.
- The ethics approval date (2019) is well after the inclusion of patients (2004 onward). The authors should clarify how retrospective data collected before 2019 were ethically covered.
Author Response
Replies to REVIEWER 3
This manuscript addresses the association between serum 25(OH)-vitamin D levels and cardiac involvement in children with acute rheumatic fever (ARF). The topic is of clinical and epidemiological interest, given the ongoing burden of rheumatic heart disease (RHD) globally, especially in low- and middle-income countries. The authors present a single-center retrospective analysis spanning two decades. The findings—suggesting that hypovitaminosis D is independently associated with the development and severity of RHD—are provocative and may open avenues for preventive or adjunctive strategies. The manuscript is clearly written and generally well-structured. However, there are several methodological limitations, interpretative overstatements, and presentation issues that need to be addressed before the work can be considered for publication. Below you will find my comments divided into major and minor. Major: The retrospective design and single-center nature of the study are major limitations. Although the authors acknowledge this, the conclusion occasionally overstates the causal relationship between vitamin D and RHD. The text should consistently frame the results as associations, not as evidence of causality.
My coauthors and I thank you so much the reviewer for her/his comments and for the opportunity of amending some inappropriate sentences within the paper.
Nutritional status, obesity, outdoor activity, and socioeconomic determinants are all strong predictors of vitamin D deficiency in children. None of these variables were collected or adjusted for. Their omission raises concerns about residual confounding. This should be more thoroughly addressed in the discussion.
This is the most important limitation of our study, which has been transparently and frankly admitted. The study starts from the retrospective assessment of vitamin D levels in a cohort of pediatric patients over two decades: at the start it was not programmed to specifically consider vitamin D status in such patients.
The cohort includes only 78 patients over 20 years, exclusively Caucasian, from a single Italian center. This restricts external validity, particularly as ARF and RHD are more prevalent in non-Caucasian populations. The limitations in terms of generalizability should be discussed more explicitly.
We did it, hoping that our frank declarations within the discussion might be considered shareable.
The multivariable logistic regression (Table 2) shows an extremely wide confidence interval for vitamin D (<30 ng/mL, OR ~27, CI 2.9–267). This suggests instability of the model due to the limited sample size and event distribution. The authors should discuss this limitation more carefully and avoid describing the results as “outstanding diagnostic accuracy” without nuance.
We have omitted the adjective “outstanding”, which was used to judge the diagnostic accuracy. We have added a sentence in the text at page 6: “The results of the logistic regression have been shown in Table 2, revealing a wide confidence interval that may be related to instability of our model, likely explained by the relatively limited sample size of patients”.
The ROC AUC of 0.965 and a cut-off of 32.5 ng/mL seem implausibly high in a real-world clinical setting. There is a risk of overfitting. The authors should test internal validation methods (e.g., bootstrapping) or, at the very least, tone down the strength of their claims.
We have regenerated the ROC curve with 5,000 bootstrap replications to make data more robust. However, we have also mitigated our considerations, of course.
All patients were said to have at least 5 years of follow-up, but some “recently diagnosed cases” were also included. This inconsistency should be clarified.
The majority of our patients had a very long follow-up, with a few exceptions who - however - were followed-up for a minimal period of at least 5 years. For patients diagnosed from 2020 to 2024 the follow-up has been evidently shorter.
The definition of “moderate/severe” RHD is heterogeneous (valvular regurgitation vs. heart failure). Please justify why these categories were merged and discuss how this might bias results.
You are perfectly right; however, the pediatric cardiologists who collaborated to this paper (GDR, ABD, CDP) were actively involved in the process of severity definition for valvular changes. More specifically, the severity of both mitral and aortic regurgitations was judged via echocardiography with metrics related to jet length ≥2 cm and peak velocity >3 m/s. This consideration has been added within the text in the section of Results. The same pediatric cardiologists were also involved in the management of cases with heart failure.
The discussion includes several citations on vitamin D and cardiovascular/autoimmune disease, but the novelty of this study relative to prior work is not sufficiently emphasized. The authors should highlight what new information is contributed compared to previous studies in Nepal, India, and the Middle East that already reported vitamin D deficiency in RHD.
There are references dedicated to Nepalese, Indian, Turkish, Iranian and Mexican patients within the discussion, mostly suggesting an association between reduced levels of serum vitamin D and different pictures of ARF-related heart diseases.
These are the Minor comments: The abstract should state explicitly that this is a retrospective, single-center study, to avoid misleading readers.
Done. Even the title of the paper declares the single-centre structure of the study.
The results section of the abstract should include actual odds ratios and confidence intervals rather than only qualitative statements.
Included.
Table 1 mixes means with medians/IQRs in a confusing way. Please use consistent reporting formats.
We have tried to explain in the statistical section that data were presented as median and interquartile range for not normally distributed continuous variables and with mean and standard deviations for normally distributed continuous variables. For this reason we have used all these terms in Table 1.
Some p-values (e.g., 0.0001) should be reported as “<0.001” according to standard conventions.
Done.
The ROC curve (Figure 1) should include the 95% CI for AUC on the figure itself.
A ROC curve with 5000 bootstrap replications was generated to obtain the 95% CI. These data have been included within figure 1.
Several expressions are overly strong (e.g., “outstanding accuracy,” “only factor associated”). Replace with more cautious wording.
Done.
Minor grammatical issues (e.g., “geoepidemiological variations… remains to unravel”) should be corrected.
Done.
The ethics approval date (2019) is well after the inclusion of patients (2004 onward). The authors should clarify how retrospective data collected before 2019 were ethically covered.
Ours is a retrospective study, and the inclusion of patients previously diagnosed with ARF (from 2004 to 2019) was accordingly performed. The local ethical committee also approved through a data protection impact assessment our study extension to include patients with ARF who were diagnosed more recently (i.e. after 2019).
Round 2
Reviewer 1 Report
Comments and Suggestions for Authors
The authors have addressed all comments. Revised form of the manuscript is suitable for being published.
Author Response
Thank you so much,
DR & coauthors
Reviewer 2 Report
Comments and Suggestions for Authors
While my comments have been addressed, the introduction still requires more focus. Talking about groups A, B, C, E. F, G etc doesn't relate to the study. It would be more useful to report what the rates of ARF/RHD are in Italy and why the need to focus on this topic.
The methods details are still missing. For example, having the questionnaire as a supplement would be useful. They say out of 90 eligible patients, but don't say how they are eligible.
Author Response
Thank you so much for your kind feedback: we have omitted in the Introduction the part referring to groups A, B, C and so on of streptococcal species.
The exact prevalence of ARF/RHD is unfortunately not known in Italy (there are only data limited to some regions in Italy, but not data expandable to the whole nation).
Pediatric patients for this study were initially 90, all with a post-streptococcal illness, but 11 were not considered as shown to be affected by post-streptococcal arthritis (which cannot be considered ARF), while 1 was excluded due to incomplete inpatient data available.
A simple questionnaire was focused on the eventual exposure to risk factors during the four weeks prior to illness and was administered at time of the first clinical assessment.
We have highlighted these small amendments within the text, thank you again for the opportunity,
DR & coauthors
Reviewer 3 Report
Comments and Suggestions for Authors
Dear Authors,
Despite the inherent limitations of a single-center retrospective study with a modest sample size, the current manuscript provides valuable exploratory insights into the relationship between vitamin D deficiency and cardiac involvement in ARF. The transparency with which you now present the study’s constraints reinforces the credibility of your findings. I encourage you to maintain a cautious interpretation throughout, particularly in the conclusions, so that the exploratory nature of the work remains evident to the reader.
Author Response
Thank you again for highlighting the cautious interpretation of our results, mostly within the conclusive remarks.
We have added a sentence in the conclusions ("Further work should warrant investigations to define the role of vitamin in ARF and RHD pathophysiology and establish the potential existence of temporal relationships between vitamin D status and disease onset, activity, and treatment response on larger cohorts of patients with a standardized assessment of potential confounding factors as diet, sun exposure, ethnicities and medications"),
DR & coauthors